# Can the Atherogenic Index of Plasma (AIP) Be a Prognostic Marker for Good Clinical Outcome After Mechanical Thrombectomy for Acute Ischemic Stroke?

**DOI:** 10.3390/diagnostics15080947

**Published:** 2025-04-08

**Authors:** Sena Boncuk Ulaş, Türkan Acar, Halil Alper Eryılmaz, Esra Ünal, Yeşim Güzey Aras, Eren Kılıç, Hakan Saçlı, Salih Salihi, Bilgehan Atılgan Acar

**Affiliations:** 1Independent Researcher, 59140 Dunkerque, France; senaboncuk@gmail.com; 2Department of Neurology, Faculty of Medicine, Sakarya University, Sakarya 54100, Turkey; turkanacar@sakarya.edu.tr (T.A.); yesimguzeyaras@hotmail.com (Y.G.A.); 3Department of Neurology, Sakarya Training and Research Hospital, Sakarya 54100, Turkey; dr.alpereryilmaz@gmail.com (H.A.E.); esrayazici95@gmail.com (E.Ü.); erenkilicmd@gmail.com (E.K.); 4Department of Cardiovascular Surgery, Faculty of Medicine, Sakarya University, Sakarya 54100, Turkey; mdhakans@yahoo.com (H.S.); drssalihi@yahoo.com (S.S.)

**Keywords:** acute stroke, mechanical thrombectomy, dyslipidemia, Atherogenic Index of Plasma

## Abstract

**Background**: Stroke remains a leading cause of morbidity and mortality worldwide, with dyslipidemia playing a crucial role in atherosclerosis and stroke development. The Atherogenic Index of Plasma (AIP), calculated as log(triglyceride/HDL), has emerged as a biomarker for atherosclerosis and cardiovascular risk. However, its relationship with stroke prognosis remains unclear. This study aimed to investigate the association between AIP and favorable clinical outcomes at three months in acute ischemic stroke patients undergoing mechanical thrombectomy. **Methods**: We conducted a retrospective analysis of 222 patients who underwent mechanical thrombectomy between December 2019 and April 2023. The association between AIP and demographic variables, etiology, successful recanalization, intracerebral hemorrhage, and three-month mRS was evaluated. AIP values were compared between patients with good (mRS 0–2) and poor (mRS 3–6) clinical outcomes. **Results**: The most common comorbidity was hypertension (72.1%), followed by AF (50%). Stroke etiologies included large artery atherosclerosis (16.2%), cardioembolism (57.2%), and undetermined causes (26.6%). AIP values were significantly lower in patients with good functional outcomes. Additionally, AIP values were inversely associated with AF but positively correlated with DM and previous stroke history. No significant relationship was observed between the AIP and successful recanalization or intracerebral hemorrhage. **Conclusions**: This study is the first to demonstrate that elevated AIP is associated with poor functional outcomes after three months in patients undergoing mechanical thrombectomy. Given its strong correlation with prognosis, the AIP may serve as a valuable biomarker for identifying high-risk patients. Future prospective studies are needed to further validate these findings and explore the potential role of the AIP in stroke management.

## 1. Introduction

Stroke is one of the leading causes of morbidity and mortality worldwide and remains a significant global health challenge. Although stroke-related mortality has declined substantially in recent years, improvements in the prognosis of acute stroke patients may still be insufficient [1,2]. Data from the 2019 Global Burden of Stroke report indicate that without urgent intervention, stroke incidence and associated mortality and morbidity will continue to rise, particularly in low-income countries [3]. Therefore, early identification of individuals at high risk of stroke and those likely to have a poor prognosis is of critical importance [1,4].

Ischemic strokes account for 85% of all acute strokes, with atherosclerosis being one of the most significant risk factors. Cardiovascular risk factors such as diabetes mellitus, dyslipidemia, and atrial arrhythmias including atrial fibrillation increase with age and contribute to panvascular atherosclerosis. In individuals over 70 years of age, the prevalence of diabetes exceeds 35%, often coexisting with lipid abnormalities and atrial arrhythmias. This constellation of risk factors predisposes to adverse cerebrovascular outcomes, including ischemic stroke [5]. Therefore, a careful prevention and monitoring strategy is crucial, particularly in elderly patients undergoing mechanical thrombectomy. Dyslipidemia plays a key role in the development of atherosclerosis [1,4,6,7]. Various blood lipid parameters have been used to assess stroke risk and predict prognosis [6,8,9]. Dyslipidemia, including triglycerides (TGs), total cholesterol (TC), low-density lipoprotein cholesterol (LDL-C), and non-high-density lipoprotein cholesterol (non-HDL-C) concentrations, increases the risk of atherosclerosis development and ischemic stroke [4]. Although lipid abnormalities are more prominent in stroke cases caused by large artery atherosclerosis, they are also significant risk factors for ischemic stroke in general [6]. However, the prognostic and predictive value of blood lipids and related markers remains limited [1]. The Atherogenic Index of Plasma (AIP) was initially introduced as a biomarker for atherosclerosis and is calculated as the logarithm of the ratio of serum TG to HDL ([log (TG/HDL)]). Previous studies indicate that a high Atherogenic Index of Plasma (AIP) is positively associated with the risk of cardiovascular disease (CVD) and may serve as a strong biomarker for predicting cardiovascular events. Furthermore, recent research highlights the importance of non-traditional lipid parameters in the development of atherosclerosis and their close association with the clinical outcomes of acute ischemic stroke [4,9,10]. However, the relationship between the AIP and acute ischemic stroke remains unclear due to limited research on its potential link to stroke risk and prognosis.

Given the potential role of the AIP as a significant biomarker for dyslipidemia and cardiovascular diseases, it may also have important implications in stroke pathogenesis. However, studies investigating the association between the AIP and acute ischemic stroke are limited. In particular, the prognostic value of the AIP in patients undergoing mechanical thrombectomy remains unclear. Mechanical thrombectomy has revolutionized the treatment of acute ischemic stroke caused by large vessel occlusion, offering significant clinical benefits when performed promptly and effectively. Over the past decade, evolving techniques and growing evidence have expanded its indications and refined patient selection strategies, significantly improving functional outcomes in selected cases [11]. Mechanical thrombectomy is the standard of care for patients with acute ischemic stroke due to large vessel occlusion. These patients typically present with more severe strokes and higher baseline disability, making prognostic biomarkers particularly valuable in predicting recovery and guiding rehabilitation efforts. Investigating the AIP in this specific cohort helps to identify whether atherogenic dyslipidemia impacts recovery despite successful recanalization, thus offering clinical insight into metabolic contributions to post-thrombectomy prognosis.

Despite advances in acute stroke treatment, a considerable proportion of patients experience unfavorable outcomes even after successful mechanical thrombectomy. Several mechanisms have been proposed to explain this phenomenon, including reperfusion injury, distal microembolization, and microvascular dysfunction. However, these factors do not fully account for the variability in clinical outcomes. We hypothesize that underlying systemic vascular conditions—such as atherogenic dyslipidemia represented by elevated AIP—may play a role in this paradox. Investigating the AIP in this specific cohort may provide insights into metabolic contributors that affect recovery independently of procedural success. This study aims to explore the relationship between AIP levels and three-month clinical outcomes in acute ischemic stroke patients treated with mechanical thrombectomy.

## 2. Materials and Methods

For this study, the data of patients who underwent mechanical thrombectomy between December 2019 and April 2023 at the Stroke Unit of the Neurology Clinic, Sakarya University Training and Research Hospital, were retrospectively reviewed. Prior to initiating the study, ethics committee approval was obtained from our university’s ethics committee (Approval No: E-71522473-050.01.04-267867-226 and Approval Date: 28 July 2023).

Demographic data—including age, gender, and comorbid conditions such as hypertension (HT), diabetes mellitus (DM), atrial fibrillation (AF), coronary artery disease, previous stroke, history of smoking, alcohol use, and obesity—were collected from patient records. Additional data on intravenous tissue plasminogen activator (IV-tPA) administration, as well as antiaggregant and anticoagulant therapies, were also recorded. The data regarding prior use of antiplatelet and anticoagulant medications were obtained from patients’ medical records and recorded accordingly. Clinical parameters such as National Institutes of Health Stroke Scale (NIHSS) [12] scores at hospital admission and at 24 h, symptom-to-door and symptom-to-recanalization times, stroke laterality (right hemisphere, left hemisphere, posterior circulation), and Alberta Stroke Program Early CT Scores (ASPECTS) [13] from brain computed tomography (CT) were retrospectively analyzed. The presence of acute internal carotid artery (ICA) occlusion, successful recanalization, first pass recanalization, intracerebral hemorrhage (ICH) (according to the Heidelberg bleeding classification) [14] and its subtypes, and modified Rankin Scale (mRS) scores at three months were also recorded from patient records.

Patients with an indication for mechanical thrombectomy were categorized into three groups based on the TOAST classification [15]: large artery atherosclerosis, cardioembolic stroke, and undetermined etiology. Strokes with negative evaluations or multiple possible etiologic causes were classified as undetermined. Since small vessel disease is not an indication for mechanical thrombectomy, these patients were excluded from the study. Additionally, two patients with stroke due to COVID-19 pneumonia—a rare cause of stroke—and four patients who presented with acute stroke secondary to carotid artery dissection and underwent mechanical thrombectomy were excluded. Furthermore, 16 patients who were already receiving lipid-lowering therapy were excluded, leaving a total of 222 patients in the final analysis.

AIP was calculated as log (TG/HDL) using TG and HDL values obtained from blood test results within the first 24 h. The correlation between AIP and various clinical and demographic factors—including age, gender, comorbidities, successful recanalization, ICH within the first 24 h, symptomatic ICH, ICA occlusion, stroke etiology, and three-month mRS—was analyzed. Successful recanalization was assessed using the Thrombolysis in Cerebral Infarction (TICI) [16] scale. The modified TICI (mTICI) classification was used, with mTICI 2c and mTICI 3 defined as successful recanalization. mTICI 2c represents near-complete perfusion, except for slow flow or a few distal cortical emboli, while mTICI 3 indicates complete perfusion [17].

In the descriptive statistics of the data, mean, standard deviation, median, lowest, highest, frequency, and ratio values were used. The distribution of variables was measured with the Kolmogorov–Smirnov test. Independent sample *t*-test and Mann–Whitney U test were used in the analysis of independent quantitative data. The Chi-square test was used in the study of categorical data, and the Fischer test was used when the Chi-square test conditions were not met. Pearson and Spearman correlation analyses were used in the correlation analysis. A multiple linear regression analysis was performed; the multiple linear regression analysis assesses the influence of all independent variables analyzed on the AIP value. For further analysis, the variable inclusion criteria into the statistical model was a significance level of 0.05. A significance of 0.05 was used for all statistical tests performed in this study. The SPSS 27.0 program was used in the study.

## 3. Results

Among the cases, 50.9% were female, with a mean age of 67.70 ± 11.20 years. The mean NIHSS score at admission was 16.88 ± 4.00, while the mean 24 h NIHSS score was 11.81 ± 10.64. IV-tPA treatment was administered to 16.2% of the cases before mechanical thrombectomy. The most common comorbidity was HT, present in 72.1% of patients, followed by AF, which was identified in 50% of cases. A history of previous stroke was present in 20.7% of the cases. The mean HDL and TG levels were 42.45 ± 10.90 mg/dL and 110.98 ± 99.54 mg/dL, respectively. The mean AIP value for all cases was 0.34 ± 0.32. Demographic characteristics, NIHSS scores, medical history, comorbidities, lipid parameters, and AIP values are summarized in Table 1.

Regarding stroke localization, 45.5% of cases involved the right hemisphere, 48.2% affected the left hemisphere, and 6.3% involved the posterior circulation. The mean ASPECT score was 9.12 ± 0.93. The mean symptom-to-door time was 117.82 ± 97.63 min, while the mean symptom-to-recanalization time was 256.45 ± 115.18 min. ICA occlusion was detected in 29.7% of cases, including 17 cases of ICA–middle cerebral artery (MCA) tandem occlusions and 49 cases of distal ICA occlusions. Among the 14 cases of posterior circulation strokes, 6 had more than 50% atherosclerosis in the basilar and/or vertebral arteries, while the remaining cases had a cardioembolic etiology.

Successful recanalization (mTICI 2c–3) was achieved in 80.2% of cases, with first-pass successful recanalization in 46.4% of cases. ICH was detected on CT scans within 24 h post-thrombectomy in 51 patients (23%). Among the ICH subtypes, type 2 parenchymal hematoma was the most common, observed in 7.2% of patients, followed by type 1 parenchymal hematoma in 6.3%, and 12% of the cases were symptomatic ICH. Stroke localization, neuroimaging findings, Digital Subtraction Angiography (DSA) characteristics, ICH types, stroke etiology, and outcome data are detailed in Table 2.

The mean mRS score at three months was 2.58 ± 2.41. A good clinical outcome (mRS 0–2) at three months was observed in 49.1% of cases. Regarding stroke etiology, 16.2% of cases were attributed to large artery atherosclerosis, 57.2% to cardioembolism, and 26.6% to undetermined causes (Table 2). In the undetermined subgroup, approximately half of the cases had multiple etiological factors.

The comparison of AIP values with NIHSS at admission, gender, ICA occlusion, medical history, successful recanalization, ICH, and 3-month mRS is presented in Table 3. AIP values were found to be significantly higher in patients with an admission NIHSS score of 10 or above (*p*: 0.032). AIP values were significantly lower in patients with AF compared to those without AF (*p*: 0.005). Conversely, AIP values were significantly higher in patients with DM compared to those without (*p*: 0.028). However, no statistically significant association was found between AIP and other comorbid conditions. Additionally, there was no significant difference in AIP values concerning successful recanalization or ICH. When we examined the outcomes, patients with a 3-month mRS of 0–2 had significantly lower AIP values compared to those with poor outcomes (mRS 3–6) (*p*: 0.007) (Table 3) (Figure 1).

When stroke cases were categorized into three etiological groups—large artery atherosclerosis, cardioembolic, and undetermined—no significant difference in AIP values was observed between the subgroups. Additionally, when we examined the relationship between AIP values and good versus poor clinical outcomes in each subgroup, it was observed that AIP values were higher in the groups with poorer clinical outcomes across all subgroups. However, this relationship was found to be statistically significant only in the undetermined etiology subgroup (*p*: 0.02) (Table 4) (Figure 2).

A multiple linear regression analysis was conducted to examine the relationship between AIP and all variables assessed in the study (Table 1 and Table 2). This analysis identified AF, previous stroke history, and 3-month mRS as factors significantly associated with AIP. The relationship between AF and AIP was negative, whereas previous stroke history and three-month mRS showed a positive association with AIP. The statistically significant associations from the regression analysis are summarized in Table 5.

## 4. Discussion

Stroke is the second leading cause of mortality and the third leading cause of morbidity worldwide. High cholesterol levels and dyslipidemia are strongly associated with stroke, and managing dyslipidemia has been shown to reduce stroke risk. Numerous studies have demonstrated the relationship between blood lipid levels and stroke [18,19,20]. Low-density lipoprotein (LDL) cholesterol is the most commonly used lipid parameter for predicting stroke risk, while high levels of HDL cholesterol are known to have a protective effect. TGs are considered markers of increased residual cholesterol particles in the blood, which contribute to atherosclerosis and atherothrombosis [21]. Research indicates that the AIP serves as a biomarker for both atherosclerosis and dyslipidemia and is positively correlated with cardiovascular disease risk. Moreover, the AIP alone has been shown to have a stronger association with cerebrovascular disease risk compared to other cholesterol parameters [1,22,23,24]. Data on the significance of the AIP in acute ischemic stroke patients undergoing mechanical thrombectomy are quite limited. The key finding of our study is that higher AIP values are associated with poor clinical outcomes in this patient group.

In a study published in 2024 investigating lipid profiles in patients with acute ischemic stroke, 50% of the cases were female, with a mean age of 71.18 ± 11.92 years [8]. Another study examining AIP values in acute ischemic stroke patients undergoing mechanical thrombectomy reported a mean age of 70.2 ± 12.1 years, with 62.1% of the patients being male [25]. In our study, 50.9% of the patients were female, and the mean age was 67.70 ± 11.20 years, which is consistent with the findings in the literature.

When examining the comorbidities of the cases, a study published in 2024 that investigated AIP values in acute ischemic stroke patients undergoing mechanical thrombectomy reported HT as the most common comorbidity, affecting 70.7% of patients [25]. Similarly, in our study, the most common comorbidity was also HT, found in 72.7% of the patients. The same study reported a median ASPECT score of 9 and a median baseline NIHSS score of 13 for the patients [25]. Another study, which also focused on acute ischemic stroke patients, identified HT as the most frequent comorbidity but reported a mean baseline NIHSS score of 3. However, unlike our study, this research included all acute ischemic stroke patients rather than those undergoing mechanical thrombectomy [1]. In our study, the mean ASPECT score was 9.12 ± 0.93, while the mean admission NIHSS score and 24 h NIHSS score were 16.88 ± 4.00 and 11.81 ± 10.64, respectively. The relatively higher NIHSS scores observed in our cohort suggest a more severe clinical presentation at the time of admission, possibly reflecting the selection criteria for patients undergoing mechanical thrombectomy.

In evaluating good clinical outcomes—the primary endpoint of our study—high AIP values were found to be associated with poorer outcomes in acute stroke patients who underwent mechanical thrombectomy. Patients with a 3-month mRS score between 0 and 2 had significantly lower AIP values. A prospective study in the literature similarly found that high AIP values were associated with an increased risk of ischemic stroke [26]. Additionally, a 2021 study involving 1463 acute stroke patients reported that elevated AIP values were linked to poorer prognosis, with this association being particularly pronounced in the large artery atherosclerosis subgroup [1]. The only study in the literature that examined the AIP in acute ischemic stroke patients undergoing mechanical thrombectomy [25] found that high AIP values were associated with early neurological deterioration. However, unlike our study, this research did not evaluate clinical outcomes based on 3-month mRS scores but focused solely on early neurological deterioration within the first 24 h. Our study is the only one to examine the relationship between the AIP and clinical outcomes at the third month in patients undergoing mechanical thrombectomy.

Another finding in our study is that AIP values were significantly lower in patients with AF compared to those without AF. The relationship between AF and dyslipidemia remains highly debated. Several studies have shown that low HDL cholesterol levels contribute to AF development, particularly through mechanisms such as increased left ventricular mass, diastolic dysfunction, and heart failure, ultimately leading to left atrial remodeling [27,28,29,30]. The relationship between other lipid parameters and AF is even more complex. Some studies have reported an inverse relationship between total cholesterol and LDL cholesterol levels and AF, while others have found that TG levels are lower in patients with AF [27,31,32]. These conflicting findings regarding AF and lipid levels are often referred to as the “paradox of dyslipidemia”, suggesting that AF may be associated with lower TG and cholesterol levels [27]. Our findings can also be interpreted as contributing to this paradox, as the significantly lower AIP values in AF patients in our study align with the complex relationship between lipid profiles and AF.

In our study, AIP values were significantly higher in patients with DM compared to those without DM. It is well established that patients with DM have a higher risk of dyslipidemia and atherogenic cardiovascular disease. These patients typically exhibit high TG and LDL cholesterol levels, along with low HDL cholesterol levels [33]. A meta-analysis of 15 studies reported a consistent positive correlation between AIP and type 2 DM across all studies, suggesting that AIP may be an effective lipid parameter for assessing type 2 DM risk [34]. Our findings align with these results.

When we categorized strokes into three etiological subgroups—large artery atherosclerosis, cardioembolic, and undetermined—we found no statistically significant differences in AIP values among these groups. Few studies have examined AIP values according to stroke subtypes. A 2022 study classified strokes into four groups: intracranial large artery atherosclerosis, extracranial large artery atherosclerosis, cardioembolism, and cryptogenic stroke. The study found elevated AIP values in both large artery atherosclerosis subgroups, whereas a negative correlation was observed in the cardioembolic and cryptogenic stroke groups [35]. However, this study was conducted in an East Asian population, where intracranial atherosclerosis is more prevalent [35,36]. In our study, intracranial atherosclerosis was a rare cause of stroke, with only eight cases in the large artery atherosclerosis group. Although we found a negative correlation between AF and AIP, this relationship was not observed in the cardioembolic stroke subgroup. This outcome was expected, as atherosclerosis and dyslipidemia are known to play significant roles in the pathophysiology of cardioembolic strokes [27]. Furthermore, in our dataset, AF accounted for 87% of the cardioembolic stroke subgroup, which also included patients with heart valve replacement, cardiac thrombus, cardiomyopathy, and recent myocardial infarction (within the last four weeks). Therefore, the observed association between AF and AIP may not fully reflect the characteristics of the cardioembolic stroke subgroup.

In a 2024 study, the AIP was found to be associated with early neurological deterioration, particularly in the large artery atherosclerosis subgroup, with no significant relationship observed in other stroke subtypes [25]. Similarly, in our study, we did not find a relationship between AIP values and stroke subtypes. However, when we analyzed the relationship between AIP values and clinical outcomes within each subgroup, we found that, as in the overall group, higher AIP values were associated with poorer clinical outcomes. This association was statistically significant only in the undetermined etiology subgroup. This may be due to the heterogeneous distribution of cases across subgroups in our analysis, with relatively small sample sizes in certain subgroups.

## 5. Conclusions

In conclusion, our study is the first to investigate the relationship between AIP values and 3-month clinical outcome in patients undergoing mechanical thrombectomy for acute ischemic stroke. The results suggest that elevated AIP values are significantly associated with poor clinical outcomes at the 3-month mark, with patients showing good clinical outcomes (mRS 0–2) exhibiting significantly lower AIP values. These findings position the AIP as a valuable biomarker for predicting stroke prognosis, independent of stroke subtypes. Furthermore, this study also highlights the association between AIP and comorbid conditions such as AF and DM, reinforcing its potential as an effective parameter for assessing stroke risk in these patients. While we did not observe statistically significant differences in mean AIP values between the three main stroke subtypes, Table 4 indicates that elevated AIP values were consistently associated with poorer outcomes within each subgroup, reaching statistical significance in the undetermined etiology group. These findings suggest that the AIP may serve as a prognostic marker across different stroke subtypes, though its predictive value may vary depending on etiology. Ultimately, our study emphasizes the AIP’s clinical significance, suggesting its role as an important predictor of outcomes in acute ischemic stroke patients undergoing mechanical thrombectomy, with implications for stroke recurrence prediction.

Our study has several limitations that should be considered when interpreting the results. Firstly, the unequal distribution of cases across stroke subtypes and relatively small sample sizes in certain subgroups may have reduced the power to detect significant differences between these groups. Additionally, this study’s cross-sectional design limits the ability to draw causal inferences, and further longitudinal studies are needed to explore the long-term prognostic value of the AIP. Despite these limitations, our study provides valuable insights into the relationship between AIP and stroke outcomes, laying the foundation for future research to confirm and expand upon these findings.

## Figures and Tables

**Figure 1 diagnostics-15-00947-f001:**
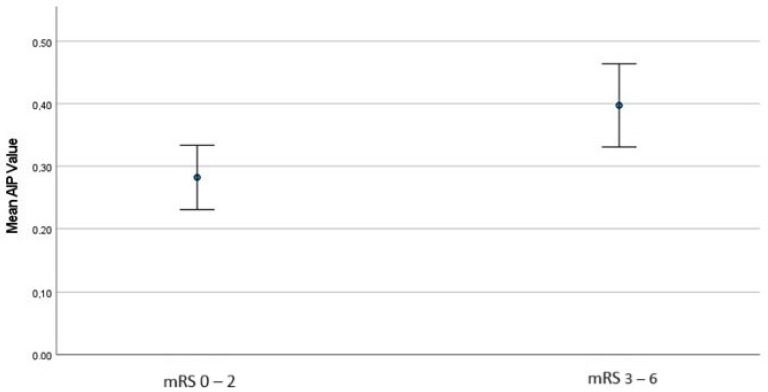
The comparison of AIP values between good and poor outcome groups.

**Figure 2 diagnostics-15-00947-f002:**
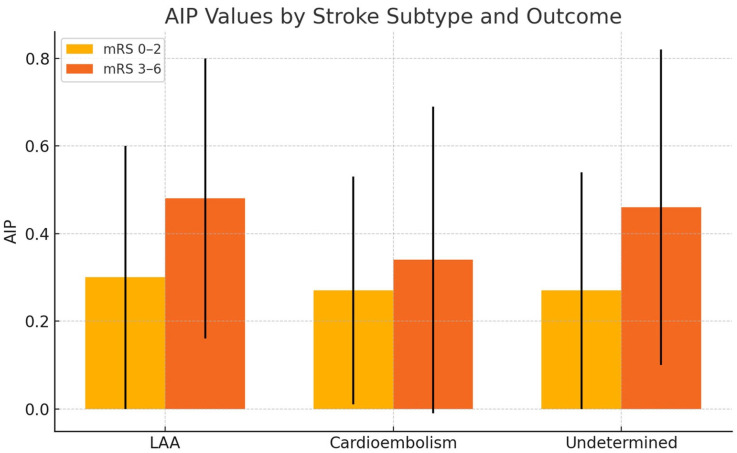
The comparison of mean AIP values between patients with good (mRS 0–2) and poor (mRS 3–6) clinical outcomes at 3 months, across different stroke subtypes.

**Table 1 diagnostics-15-00947-t001:** Demographic data, NIHSS, medical history, HDL, triglyceride, and AIP values of the cases.

		Min–Max	Median	*n*	%
Sex	Female			113	50.9%
	Male			109	49.1%
Age		29–94	69.00		67.70 ± 11.20
Admission NIHSS		4–38	17.00		16.88 ± 4.00
24 h NIHSS		0–42	11.00		11.81 ± 10.64
IV tPA (+)				36	16.2%
Atrial fibrillation (+)				111	50.00%
Hypertension (+)				160	72.1%
Active smoking (+)				57	25.7%
Ex-smoking (+)				20	9.00%
Alcoholism (+)				21	9.5%
Obesity (+)				40	18%
DM (+)				52	23.4%
CABG (+)				13	5.9%
CAD (+)				80	36.00%
MI in the last 3 months (+)				3	1.4%
Previous stroke history (+)				46	20.7%
HDL		11–80	42		42.45 ± 10.90
Triglyceride		7–1015	84.50		110.98 ± 99.54
AIP value		−0.78–1.86	0.30		0.34 ± 0.32

Abbr: Min: Minimum, Max: Maximum, NIHSS: National Institutes of Health Stroke Scale, IV tPA: Intravenous tissue plasminogen activator, DM: Diabetes mellitus, CABG: Coronary artery bypass grafting, CAD: Coronary artery disease, MI: Myocardial infarction, HDL: high-density lipoprotein, AIP: Atherogenic Index of Plasma.

**Table 2 diagnostics-15-00947-t002:** Stroke localization, neuroimaging and DSA findings, ICH, etiology, and outcome of the cases.

			Min–Max	*n*	%
Stroke localization		Right		101	45.5%
		Left		107	48.2%
		Posterior		14	6.3%
ASPECTS			6–10		9.12 ± 0.93
Symptom-to-door time (min)			0–571		117.82 ± 97.63
Symptom-to-recanalization time (min)			17–878		256.45 ± 115.18
ICA occlusion		(+)		66	29.7%
		(−)		156	70.3%
Successful recanalization (mTICI2c-3)		(+)		178	80.2%
		(−)		44	19.8%
First-pass recanalization		(+)		103	46.4%
		(−)		119	53.6%
ICH at 24 h		(+)		51	23.0%
		(−)		171	77.0%
Types of ICH		Type 1 HI		8	3.6%
		Type 2 HI		6	2.7%
		Type 1 PH		14	6.3%
		Type 2 PH		16	7.2%
		Subarachnoid hemorrhage		7	3.2%
Symptomatic ICH				27	12%
Three-month mRS			0–6		2.58 ± 2.41
Cases with three-month mRS 0–2				109	49.1%
Etiology	LAA			36	16.2%
	Cardioembolism			127	57.2%
	Undetermined			59	26.6%

Abbr: Min: Minimum, Max: Maximum, ASPECTS: The Alberta stroke program early CT score, min: Minute, ICA: Internal Carotid Artery, mTICI: The modified thrombolysis in cerebral infarction, ICH: Intracerebral Hemorrhage, 24 h: 24 h, HI: Hemorrhagic Infarct, PH: Parenchymal Hematoma, mRS: The Modified Rankin Score, LAA: Large Artery Atherosclerosis.

**Table 3 diagnostics-15-00947-t003:** Comparison of the AIP with the NIHSS, gender, ICA occlusion, medical history, successful recanalization, hemorrhagic transformation, and 3-month mRS.

		AIP Index			*p*	
		Mean. ± sd	Median	Min–Max		
Admission NIHSS	<10	0.14 ± 0.28	0.14	−0.26–0.62	0.032	^t^
	≥10	0.35 ± 0.32	0.30	−0.78–1.86		
Sex	F	0.31 ± 0.27	0.26	−0.27–1.15	0.248	^t^
	M	0.36 ± 0.36	0.33	−0.78–1.86		
ICA occlusion	(−)	0.33 ± 0.31	0.28	−0.27–1.86	0.705	^t^
	(+)	0.35 ± 0.33	0.33	−0.78–1.18		
AF	(−)	0.40 ± 0.34	0.36	−0.26–1.86	0.005	^t^
	(+)	0.28 ± 0.28	0.24	−0.78–1.07		
HT	(−)	0.32 ± 0.35	0.28	−0.26–1.86	0.582	^t^
	(+)	0.34 ± 0.30	0.30	−0.78–1.34		
DM	(−)	0.31 ± 0.29	0.27	−0.78–1.15	0.028	^t^
	(+)	0.42 ± 0.39	0.34	−0.27–1.86		
CABG	(−)	0.34 ± 0.32	0.29	−0.78–1.86	0.908	^t^
	(+)	0.35 ± 0.25	0.36	−0.10–0.78		
CAD	(−)	0.34 ± 0.34	0.30	−0.78–1.86	0.612	^t^
	(+)	0.32 ± 0.28	0.29	−0.27–1.07		
MI in the last 3 months	(−)	0.33 ± 0.32	0.30	−0.78–1.86	0.562	^t^
	(+)	0.44 ± 0.34	0.35	0.16–0.83		
Previous stroke history	(−)	0.32 ± 0.28	0.29	−0.27–1.15	0.072	^t^
	(+)	0.41 ± 0.42	0.35	−0.78–1.86		
Obesity	(−)	0.32 ± 0.32	0.29	−0.78–1.86	0.188	^t^
	(+)	0.40 ± 0.30	0.34	−0.07–1.07		
Active smoking	(−)	0.32 ± 0.29	0.29	−0.78–1.18	0.148	^t^
	(+)	0.39 ± 0.38	0.35	−0.18–1.86		
Ex-smoking	(−)	0.33 ± 0.32	0.29	−0.78–1.86	0.160	^t^
	(+)	0.43 ± 0.33	0.42	−0.10–1.02		
Alcoholism	(−)	0.34 ± 0.32	0.29	−0.78–1.86	0.937	^t^
	(+)	0.34 ± 0.29	0.36	−0.15–0.83		
Successful recanalization (mTICI2c-3)	(−)	0.37 ± 0.29	0.30	−0.19–1.15	0.502	^t^
	(+)	0.33 ± 0.32	0.29	−0.78–1.86		
ICH at 24 h	(−)	0.32 ± 0.31	0.29	−0.78–1.34	0.156	^t^
	(+)	0.39 ± 0.35	0.33	−0.18–18.6		
Three-month mRS 0–2	(−)	0.39 ± 0.35	0.33	−0.78–1.86	0.007	^t^
	(+)	0.28 ± 0.27	0.25	−0.27–0.99		

^t^ Independent Samples *t*-test. Abbr: AIP: Atherogenic Index of Plasma, sd: Standard Deviation, Min: Minimum, Max: Maximum, NIHSS: National İnstitutes of Health Stroke Scale, ICA: Internal Carotid Artery, HT: Hypertension, AF: Atrial Fibrilation, DM: Diabetes Mellitus, CABG: Coronary Artery Bypass Grafting, CAD: Coronary Artery Disease, MI: Myocardial Infarction, mTICI: The Modified Thrombolysis In Cerebral Infarction, ICH: Intracerebral Hemorrhage, 24 h: 24 h, mRS: The Modified Rankin Score.

**Table 4 diagnostics-15-00947-t004:** Distribution of AIP index among groups according to stroke etiologies.

		AIP	
Stroke Subgroup	*n*	Mean ± sd	*p*
LAA	36	0.41 ± 0.32	
Cardioembolism	127	0.31 ± 0.31	0.234
Undetermined	59	0.35 ± 0.32	
LAA	mRS 0–2, *n* = 15	0.30 ± 0.30	0.10
	mRS 3–6, *n* = 21	0.48 ± 0.32	
Cardioembolism	mRS 0–2, *n* = 60	0.27 ± 0.26	0.24
	mRS 3–6, *n* = 67	0.34 ± 0.35	
Undetermined	mRS 0–2, *n* = 34	0.27 ± 0.27	0.02
	mRS 3–6, *n* = 25	0.46 ± 0.36	

Abbr: AIP: Atherogenic Index of Plasma, sd: Standard Deviation, Min: Minimum, Max: Maximum, LAA: Large Artery Atherosclerosis.

**Table 5 diagnostics-15-00947-t005:** Significant determinants for AIP index in forward multiple linear regression model.

Dependent Variable	Regression Variables	β	SE	*p*	R^2^ of the Model
AIP	Constant	0.677	0.335	0.045	0.13
	Atrial Fibrillation	−0.180	0.046	0.013	
	Previous Stroke History	0.130	0.052	0.049	
	3rd month mRS	0.406	0.022	0.014	

β: Standardized Regression Analysis Coefficients, SE: Standard Error, Constant: Estimated Intercept Value. Abbr: mRS: The Modified Rankin Score. All data evaluated in this study (data in Table 1 and Table 2) were included in the multiple linear regression analysis, but only significant (*p* < 0.05) variables are shown in this table.

## Data Availability

Due to the nature of the research and due to ethical reasons, supporting data are not available.

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
