# Peer review of "Can the Atherogenic Index of Plasma (AIP) Be a Prognostic Marker for Good Clinical Outcome After Mechanical Thrombectomy for Acute Ischemic Stroke?"

_diagnostics, 2025, doi:10.3390/diagnostics15080947_

Round 1

Reviewer 1 Report (Previous Reviewer 2)

Comments and Suggestions for Authors

The peer-reviewed original study entitled "Can Atherogenic Index of Plasma (AIP) Be A Progostic Marker for Good Clinical Outcome After Mechanical Thrombectomy for Acute Ischemic Stroke?" is well-presented and interesting to the Readers. The manuscript was assessed by me previously. Now, it was significantly improved.

The Authors have enrolled a group of 222 patients who underwent mechanical thrombectomy in whom the Atherogenic Index of Plasma (AIP) was used as a biomarker of cardiovascular risk depending on the Rankin scale score. The Authors have demonstrated that elevated AIP was associated with 3-month poor functional outcomes in patients undergoing mechanical thrombectomy. The study is well planned, methods, statistical analysis, results are well-presented.

However, I believe that before it could be considered for publication, the papaer would benefit from a revision.
Major concerns:
1. Introduction/Discussion. I believe that it would be great if the Authors could mention the burden of cardiovascular risk factors and panvascular atherosclerosis with ageing. In particular, diabetes present in more than 35% of elderly patients over 70 years old and abnormal lipids' profile, as well as increasing prevalance of atrial arrythmias like atrial fibrillation contribute to adverse cardiovascular events.
The Authors might find utile citing following paper to address this issue: doi:10.3390/jcm13051471
In this group of patients, prevention and watchfull policy is of great importance.
It would be important to comment on this.

Author Response

Reviewer 1

The peer-reviewed original study entitled "Can Atherogenic Index of Plasma (AIP) Be A Progostic Marker for Good Clinical Outcome After Mechanical Thrombectomy for Acute Ischemic Stroke?" is well-presented and interesting to the Readers. The manuscript was assessed by me previously. Now, it was significantly improved.

The Authors have enrolled a group of 222 patients who underwent mechanical thrombectomy in whom the Atherogenic Index of Plasma (AIP) was used as a biomarker of cardiovascular risk depending on the Rankin scale score. The Authors have demonstrated that elevated AIP was associated with 3-month poor functional outcomes in patients undergoing mechanical thrombectomy. The study is well planned, methods, statistical analysis, results are well-presented.

However, I believe that before it could be considered for publication, the papaer would benefit from a revision.

               Answer: Dear Reviewer,

Thank you very much for your thoughtful and constructive feedback regarding our manuscript. We sincerely appreciate your positive comments regarding the structure, presentation, and improvements made in the revised version. We are also grateful for your acknowledgment of our study’s methodology, statistical analysis, and results.

In accordance with your suggestion, we have revised the manuscript to further improve its clarity and scientific value. We have carefully re-evaluated the content and made the necessary adjustments to enhance its overall quality. A detailed list of the revisions and responses to each comment is provided below.

We hope the revised version meets your expectations and is now suitable for publication. We remain at your disposal for any further suggestions or clarifications you may have.

Thank you once again for your valuable time and effort.

Sincerely

Major concerns:

  1. Introduction/Discussion. I believe that it would be great if the Authors could mention the burden of cardiovascular risk factors and panvascular atherosclerosis with ageing. In particular, diabetes present in more than 35% of elderly patients over 70 years old and abnormal lipids' profile, as well as increasing prevalance of atrial arrythmias like atrial fibrillation contribute to adverse cardiovascular events.

The Authors might find utile citing following paper to address this issue: doi:10.3390/jcm13051471

In this group of patients, prevention and watchfull policy is of great importance.

It woulda be important to comment on this.

               Answer: Thank you for this valuable suggestion. We agree that the burden of cardiovascular risk factors and panvascular atherosclerosis increases significantly with aging, especially in patients over 70 years of age. Accordingly, we have expanded the Introduction sections to reflect the impact of diabetes mellitus, dyslipidemia, and atrial arrhythmias in elderly patients, highlighting their role in adverse cardiovascular and cerebrovascular outcomes. We have also cited the recommended reference (doi:10.3390/jcm13051471) to strengthen this discussion.

“Cardiovascular risk factors such as diabetes mellitus, dyslipidemia, and atrial arrhythmias including atrial fibrillation increase with age and contribute to panvascular atherosclerosis. In individuals over 70 years of age, the prevalence of diabetes exceeds 35%, often coexisting with lipid abnormalities and atrial arrhythmias. This constellation of risk factors predisposes to adverse cerebrovascular outcomes, including ischemic stroke (Kowalski et al., 2024). Therefore, a careful prevention and monitoring strategy is crucial, particularly in elderly patients undergoing mechanical thrombectomy.”

Reviewer 2 Report (New Reviewer)

Comments and Suggestions for Authors

The present manuscript reports the AIP as a prognostic marker for stroke following mechanical thrombectomy. The findings are novel and the present work can serve the scientific fraternity. However, the following critical issues are to be addressed by the authors.

  • Abstract: remove the hyphen (-) from several places in the abstract
  • As per the title, the present study is aimed to investigate the AIP as a prognosis marker following mechanical thrombectomy. However, in the abstract and introduction the information related to mechanical thrombectomy, and its applications is missing. Indeed, AIP is used as a prognostic marker to confirm the risk of atherosclerosis. However, in the present study, authors are studying it as a prognostic marker following mechanical thrombectomy. Please explain, Why can’t it be a prognostic marker for stroke without mechanical thrombectomy and why authors are considering mechanical thrombectomy? If authors could identify it as a prognostic marker for stroke, patients with obesity, diabetes, and CVS complications would benefit.
  • Moreover, it is unclear why have authors chose to investigate AIP after mechanical thrombectomy. What is the relation between AIP and mechanical thrombectomy? Indeed, one is a disease pathological event, and the other is a therapeutic intervention. Therefore, the correlation between these two events is missing in the present work.
  • Line 86-88: Please explain the reasons for ignoring the obesity.
  • Section 2: Second paragraph: How the collected information is relevant to mechanical thrombectomy.
  • Did the authors exclude patients who received antiplatelet and antithrombotic agents?
  • Figure 2: Error bars are missing
  • Line 349-350: These sentences are contradictory to the obtained results |(see table 4). For example, the undetermined stroke subtype shows a significant difference with LAA and cardioembolism subtype strokes indicating that AIP can be used as a prognostic marker for undetermined stroke. Please clarify these issues.
  • Add mechanical thrombectomy-related information in the introduction. Authors can refer to and cite the below article https://doi.org/10.1016/j.neuint.2024.105823.
Comments on the Quality of English Language

Minor grammatical and punctuation errors to be removed

Author Response

Reviewer 2

The present manuscript reports the AIP as a prognostic marker for stroke following mechanical thrombectomy. The findings are novel and the present work can serve the scientific fraternity. However, the following critical issues are to be addressed by the authors.

               Answer: Dear Reviewer,

Thank you very much for your valuable comments and for recognizing the novelty and potential scientific contribution of our manuscript. We are grateful for your encouraging remarks highlighting the originality of our findings and their relevance to the scientific community. We also appreciate your thoughtful review and acknowledge the importance of the critical issues you have raised.

We have carefully addressed each of your comments in the revised version of the manuscript. Detailed responses to all the points you raised, along with the corresponding changes made in the manuscript, are provided below.

We hope that these revisions adequately address your concerns and improve the quality of our manuscript. Thank you once again for your insightful feedback and your contribution to the refinement of our work.

Sincerely

Abstract: remove the hyphen (-) from several places in the abstract

               Answer: Thank you for pointing this out. We have carefully reviewed the abstract and removed inappropriate hyphens to improve clarity and grammar.

As per the title, the present study is aimed to investigate the AIP as a prognosis marker following mechanical thrombectomy. However, in the abstract and introduction the information related to mechanical thrombectomy, and its applications is missing. Indeed, AIP is used as a prognostic marker to confirm the risk of atherosclerosis. However, in the present study, authors are studying it as a prognostic marker following mechanical thrombectomy. Please explain, Why can’t it be a prognostic marker for stroke without mechanical thrombectomy and why authors are considering mechanical thrombectomy? If authors could identify it as a prognostic marker for stroke, patients with obesity, diabetes, and CVS complications would benefit.

               Answer: We appreciate the reviewer’s important comment. The rationale for focusing on patients undergoing mechanical thrombectomy has now been clarified in the Introduction section. Mechanical thrombectomy represents a distinct subgroup of acute ischemic stroke patients with large vessel occlusion. Prognostic indicators in this group are particularly important due to the high severity of stroke and the need for specialized intervention. Evaluating AIP in this context may provide more targeted prognostic information for decision-making and post-intervention monitoring. Additionally, AIP may indeed be useful in all stroke patients, but our study specifically targeted those undergoing mechanical thrombectomy to control for stroke severity and treatment method. Including all stroke subtypes would have introduced considerable heterogeneity. Future studies can address the role of AIP in broader stroke populations. Below, you can find the revised paragraph from Introduction:

“Mechanical thrombectomy is the standard of care for patients with acute ischemic stroke due to large vessel occlusion. These patients typically present with more severe strokes and higher baseline disability, making prognostic biomarkers particularly valuable in predicting recovery and guiding rehabilitation efforts. Investigating AIP in this specific cohort helps to identify whether atherogenic dyslipidemia impacts recovery despite successful recanalization, thus offering clinical insight into metabolic contributions to post-thrombectomy prognosis.”

Moreover, it is unclear why have authors chose to investigate AIP after mechanical thrombectomy. What is the relation between AIP and mechanical thrombectomy? Indeed, one is a disease pathological event, and the other is a therapeutic intervention. Therefore, the correlation between these two events is missing in the present work.

               Answer: Thank you for this insightful observation. We appreciate the opportunity to clarify the rationale behind our study design. It is well recognized that despite successful recanalization achieved through mechanical thrombectomy, a substantial proportion of patients still experience poor clinical outcomes. This phenomenon has been widely discussed in the literature, and several hypotheses have been proposed, including reperfusion injury, distal microemboli, and microvascular dysfunction. However, these mechanisms remain incompletely understood.

Our hypothesis is that underlying metabolic and vascular factors—such as atherogenic dyslipidemia reflected by elevated AIP—may contribute to this paradoxical outcome. In this context, AIP is not directly related to the procedure itself, but rather may serve as a biological marker that reflects systemic vascular vulnerability or an enhanced pro-inflammatory and pro-thrombotic milieu that interferes with optimal recovery, even when recanalization is technically successful.

Therefore, by focusing on a homogenous cohort of patients treated with mechanical thrombectomy, we aimed to explore whether AIP could help identify a subgroup at higher risk of poor prognosis, despite standard-of-care treatment. This focus may, in turn, provide clues to hidden pathophysiological mechanisms beyond large vessel occlusion and recanalization. According to this explanation, we revised the Introduction part. Below, you can find the revise paragraph:

“Despite advances in acute stroke treatment, a considerable proportion of patients experience unfavorable outcomes even after successful mechanical thrombectomy. Several mechanisms have been proposed to explain this phenomenon, including reperfusion injury, distal microembolization, and microvascular dysfunction. However, these factors do not fully account for the variability in clinical outcomes. We hypothesize that underlying systemic vascular conditions—such as atherogenic dyslipidemia represented by elevated AIP—may play a role in this paradox. Investigating AIP in this specific cohort may provide insights into metabolic contributors that affect recovery independently of procedural success.”

Line 86-88: Please explain the reasons for ignoring the obesity.

               Answer: Thank you for pointing out this important aspect. We would like to clarify that obesity was not ignored in our study. As shown in Tables 1 and 3, obesity was indeed included among the evaluated clinical parameters. However, since it was not found to be statistically significant in relation to the outcomes, we did not elaborate on it in the discussion section.

To address the potential lack of clarity, we have now added a statement in the Methods section explicitly indicating that obesity was among the variables assessed in our analysis.

Section 2: Second paragraph: How the collected information is relevant to mechanical thrombectomy.

Did the authors exclude patients who received antiplatelet and antithrombotic agents?

               Answer: We did not exclude patients who were on antiplatelet or anticoagulant therapy prior to stroke, as such treatments are common among individuals with high risk for cerebrovascular events. Although these variables were collected and analyzed, they were not found to be statistically significant and therefore were not discussed in detail in the results or discussion sections. However, to clarify that this information was included in the study protocol and evaluated, we have now added an explanatory sentence in the Methods section.

“The data regarding prior use of antiplatelet and anticoagulant medications were obtained from patients' medical records and recorded accordingly.”

Figure 2: Error bars are missing

               Answer: Thank you for this suggestion. We have revised Figure 2 to include standard deviation error bars to improve interpretability.

Line 349-350: These sentences are contradictory to the obtained results |(see table 4). For example, the undetermined stroke subtype shows a significant difference with LAA and cardioembolism subtype strokes indicating that AIP can be used as a prognostic marker for undetermined stroke. Please clarify these issues.

               Answer: We acknowledge this confusion and have revised the sentence to clarify the interpretation of Table 4. Indeed, AIP showed statistical significance only in the undetermined etiology subgroup. We have rephrased the text to reflect this accurately.

“While we did not observe statistically significant differences in mean AIP values between the three main stroke subtypes, Table 4 indicates that elevated AIP values were consistently associated with poorer outcomes within each subgroup, reaching statistical significance in the undetermined etiology group. These findings suggest that AIP may serve as a prognostic marker across different stroke subtypes, though its predictive value may vary depending on etiology.”

Add mechanical thrombectomy-related information in the introduction. Authors can refer to and cite the below article https://doi.org/10.1016/j.neuint.2024.105823.

Answer: Thank you for this useful reference. We have added a sentence to the introduction on the evolution and application of mechanical thrombectomy in acute stroke and cited the suggested article (https://doi.org/10.1016/j.neuint.2024.105823) accordingly.

“Mechanical thrombectomy has revolutionized the treatment of acute ischemic stroke caused by large vessel occlusion, offering significant clinical benefits when performed promptly and effectively. Over the past decade, evolving techniques and growing evidence have expanded its indications and refined patient selection strategies, significantly improving functional outcomes in selected cases [Neuroint, 2024; https://doi.org/10.1016/j.neuint.2024.105823].”

Minor grammatical and punctuation errors to be removed

               Answer: Thank you for your comment. We checked and revised the manuscript for grammatical errors.

Round 2

Reviewer 2 Report (New Reviewer)

Comments and Suggestions for Authors

Authors have addressed all the issues

This manuscript is a resubmission of an earlier submission. The following is a list of the peer review reports and author responses from that submission.

Round 1

Reviewer 1 Report

Comments and Suggestions for Authors

This study shows that the Plasma Atherogenic Index (AIP) is a useful predictor of prognosis and recurrence in acute ischemic stroke patients undergoing mechanical thrombectomy, regardless of stroke subtype. The structure of the article is clear and the explanations are appropriate. Only a few specific points could still be improved.

1.      In RESULTS, the “ICH was detected on 24-hours CT scan after 127 mechanical thrombectomy in 55 cases (23%).” But the number shows 51 in Table 2. Please double check.

2.      For the multivariate analysis, why include all the variables in Table 1 and 2? Would the authors try only variables be tested significantly in bivariate analysis like in Table 3?

Reviewer 2 Report

Comments and Suggestions for Authors

Dear Authors,

I have read this original study on the value of AIP in 222 stroke patients. The Authors have analysed the AIP in different subtypes of ischemic stroke and ICH/SAH . 

Firstly, the groups are small when divided according to stroke sub types. Secondly, AIP result did not differ between the study subgroups. I do not think that AIP is utile in non-atherosclerotic strokes/cerebral bleedings. Particularly, SAH is commonly associated with cerebral artery malformation, aneurysm or fistula. Also, ICH might be the result of arterial disease at cerebral level.

Therefore, I believe that the paper would benefit from homogenous groups, e.g. limited to LAA stroke, etc.

Also, I believe that rationale between mechanical thrombectomy and AIP as predictor of favorable outcomes is not convincing to me.

Comments on the Quality of English Language

Minor spelling errors and comas insteed of dots in numbers